# Experiment on compaction of air-dried soil under drop shocks

**Jianbo Wang** [1,2], **Tiansheng Hong** [1,2], **Zhen Li** [2,3]*, **Xiuyun Xue** [2,3], **Shilei Lyu** [2,3]

**1** College of Engineering, South China Agricultural University, Guangzhou, China, **2** Division of Citrus Machinery, China Agriculture Research System, Guangzhou, China, **3** College of Electronic Engineering, South China Agricultural University, Guangzhou, China

* lizhen@scau.edu.cn

**Data Availability Statement:** All relevant data are within the paper and its Supporting Information files.

**Funding:** This work was supported by the National Natural Science Foundation of China (No. 61601189), the Special Fund of Modern

## Abstract

For the requirement in container nursery culture that growing media should be achieved the appropriate degree compaction, this paper presents an experiment on the compaction dynamics of air-dried soil under repetitive drop shocks, as a preliminary step toward the mechanization of this compaction method. The drop height used to adjust the shock intensity included 2 mm, 4 mm, 5 mm and 6 mm. And the overall packing density of soil in a vertically stratified cylinder vessel and the local packing density in each layer were taken as indicators of soil compaction states. The stretched exponential function derived from KWW law than the empirical inverse-logarithmic function has been found to be more suitable for expressing the temporal evolution of soil compaction, according to the results of curve-fitting to test values of the overall and local density. It is inherent in this experimental configuration that the drop shock intensity even at a constant drop height varies with drop times, owing to the interaction between the soil packing itself and drop shocks caused by the combination of the packing and the container. But the function $t/\tau_f(t,H)$ is manifested as a straight line on the drop times $t$ with the line slope related to the drop height $H$, so the soil compaction dynamics caused by its drop shocks and that under the condition with actively controlled intensity actually share the common relaxation law. In addition, the soil's one-dimensional distribution of local packing density showed a slight positive gradient as similar as monodisperse particles did.

## Introduction

Container seedlings have advantages [1–4] over bareroot seedlings in terms of roots intactness, survival rate and growth potential, and are especially suitable for seedling species with sensitivity to transplantation and areas with harsh site conditions [5]. Container seedlings grow in separated containers and obtain water, air, mineral nutrients and physical support from the limited medium in containers, making it a unique and critical process to fill containers with growing media [6, 7]. In despite of the diversity of growing media and containers available in the culture of container nursery, plants have similar requirements for the growing medium properties, among which the proper degree of compaction [8–11] is closely related to the filling

Technology System of Agricultural Industry (No. CARS-26), and the Characteristic Innovation Projects of Universities in Guangdong Province (No. 2018KTSCX020).

**Competing interests:** The authors have declared that no competing interests exist.

operation. Block containers [12] and non-woven fabric containers [13] are two typical cases of mechanically filling and compacting. In the former case, the automatic equipment has been designed to, in sequence, fill each cell within a block, compress the medium with indenters, sow seeds into hollows formed from the compression, and to cover these seeds by refilling. Namely, the compaction method of compressing the medium on its surface was used [14]. For the other case [13, 15], the band-shaped material of non-woven fabric is made into a continuous cylinder through the heat-sealing operation, which synchronously gets filled with the growing medium delivered and extruded by a rotating screw. And then the packed cylinder is cut into segments with required length. Using the rotating screw instead of the indenters the compaction method is actually the same as the former case. The manual operation still seems to be the only option for individual bottomed containers, such as polyethylene film and non-woven fabric bags usually used to cultivate container-grown seedlings of citrus. Operators settle the growing medium by shaking containers or tapping them on the table from experiences and usual practices [6]. This manual operation, without direct contact with the medium, is essentially different from the two mechanical ones mentioned above, allows multiple operations to execute simultaneously because the action of shaking or tapping a container could keep its opening clear. Indeed, the operation of transplanting a citrus rootstock seedling into a container requires that operators load and compact growing media alternately by one hand, while keeping the stock seedling suspended in the container with the other hand. We are motivated by these differences to make efforts to convert this manual compaction method to a mechanical integration of filling containers, compacting media and transplanting seedlings.

One should note that, however, although the method of compacting growing media by tapping containers has been applied in practice and proved to be effective, it is necessary to systematically investigate the compaction process and influence factors prior to engineering application. It is common sense that granular system could get loosed or densified subjected to external mechanical disturbance, and also a research focus in physics since at least two decades [16]. James B. Knight and others [17, 18] first considered packing fraction of the mono-sized beads packing as a function of tapping times and allowed the conclusion that the relaxation evolution was well fitted by the inverse-logarithmic law. P. Philippe et al. [19–21] carried out similar compaction experiments with a thicker tube relative to the glass beads, but found that the slow compaction of vibrated grains was better fitted by the stretched exponential function. Despite of different relaxation laws, common views [22, 23] had been reached that the dependence of the packing density on the intensity of the mechanical excitement was non-monotonic, so the packing density could be controlled within a desired scope by reasonably adjusting the intensity. In contrast to the spherical particle packing, the anisotropy particle packing [24–26] under vertical tapping evolved to a dense and highly ordered state, and the anisotropy was found to tend to amplify relaxation features like a wider range of packing density accessible. Other particle properties such as friction [27, 28], humidity [29, 30] and cohesion [16, 28] influencing interaction between particles could change initial and final packing density and characteristic times for compaction. And at times they divide the compaction process into two regimes [16, 27, 28], each of which was still well fitted to the inverse logarithmic law or the stretched exponential law. One should note that any growing medium is a poly-disperse mixture of mineral particles, organic matter, water and others, leading to much complicated particle properties.

In this work, we focus on the compaction dynamics of growing media under drop shocks, and expect to figure out whether the packing density obeys the existing relaxation laws and whether there exits any local over compaction [10, 31]. Here, we selected an inorganic soil and conducted experiments on the compaction dynamics of this material using the setup specially designed to produce successive drop shocks. The overall and local packing density were

analyzed to establish the temporal evolution and the vertical distribution, providing a preliminary experimental step towards the full exploration of the mechanically compacting growing media.

## Materials and methods

### Materials preparation

The soil material was taken at the depth more than 1 m below the ground surface in the campus of South China Agricultural University (Guangzhou, China). This type of soil, known as "Yellow Soil", is one of main components of growing media in container citrus nursery culture. In order to balance with the ambient humidity, it had been air-dried for the whole month of August, followed by being screened through 2 mm sieve pores prior to the experiment. The prepared soil weighed about 50 kg with the moisture content of 6.16% (oven-drying method [32–34]) and the relative density of 2.72 g/cm$^3$ (pycnometer method [33, 34]). Measurements of soil size grading (Fig 1) before and after the experiment were taken with particle sieving method [33, 34].

### Experimental setup and procedure

Our experimental setup included an automatic drop tester and a cylinder container matched with the tester (Fig 2). With a bottom control system based on PLC and an interactive application in PC, the setup was enabled to run and collect data automatically. This tester was designed to vertically lift the container to a specific height (called drop height, $H$) and then leave it to fall freely and dash against the rubber buffer fixed on the tester, therefore applying a shock pulse to soil in the container (Fig 2A). The intensity of drop shocks was defined as, $\Gamma$, the ratio of the peak acceleration to gravitational acceleration (Fig 2C), and measured with an accelerometer (EA-YD-152, range: 50 g, sensitivity: 101.3 mV/g) mounted at the bottom of the container. A laser ranging sensor (range: 5 m, accuracy: ±1 mm), driven by a step motor, was enabled to swing reciprocally above the container, while scanning downwards the surface profile of soil. The trajectory of the ranging sensor was an arc in a horizontal plane going through the axis of the container, and included 40 equidistant ranging points with the adjacent interval

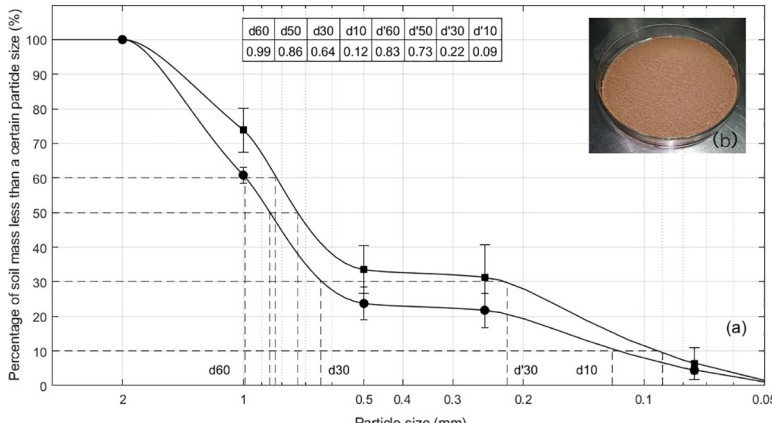

**Fig 1. Soil material and its particle size distribution.** (a) The accumulative size distribution curves of soil particles before (•) and after (■) experiments. Each set of marked points is an average of 3 separate screening operations, and error bars represent the RMS variations between operations. The solid lines are least square fits of these averages using the algorithm of shape-preserving interpolation. The table shows characteristic particle size of soil. According to observations, a large number of particle clusters formed from the cohesion of wet soil before air-drying, broke due to the experimental operations, leading to the difference between the two curves. (b) A photograph of the prepared soil.

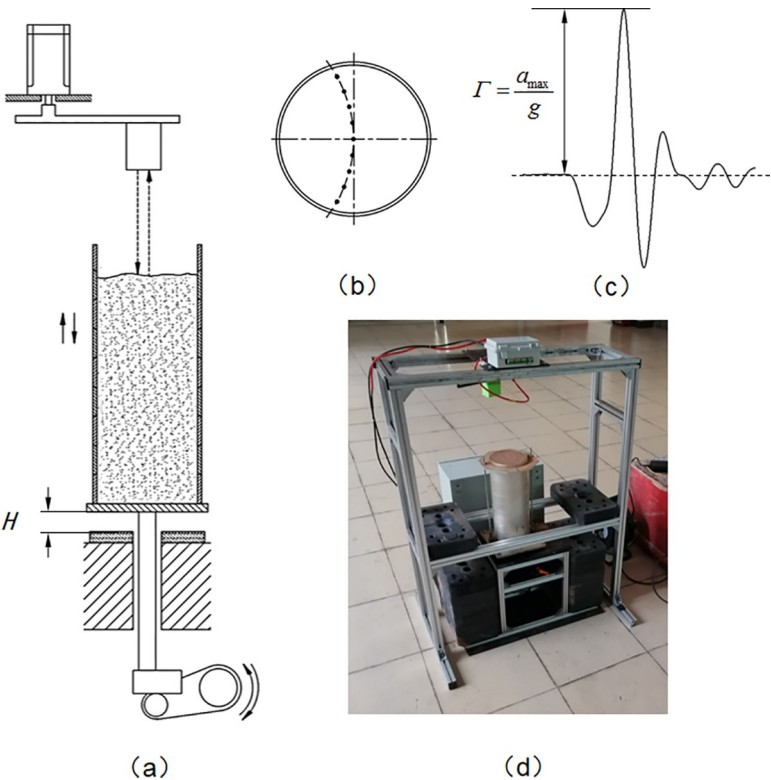

**Fig 2. The experimental setup.** (a) A schematic diagram of our experimental setup with the container installed on the drop tester, where dashed arrows indicate measuring the drop distance of soil surface using a laser ranging sensor, and the drop height, $H$, is the distance the container could be lifted. (b) The top view of ranging points marked with solid circles on soil surface. These points are evenly distributed along the arc passing through the container axis. (c) The typical acceleration waveform of a drop shock, where the intensity, $\Gamma$, is defined as the ratio of the peak acceleration, $a_{max}$, to the gravitational acceleration, $g$. (d) A picture of the apparatus prepared to run.

of 3.3 mm (Fig 2B). And the combination of the sensor and the step motor was mounted on a truss independent from the tester to isolate vibration. The container was a detachable assembly composed of a base and 10 identical aluminum-alloy tubes. These tubes were vertically stacked and formed a cylindrical space with the diameter of 13 cm and the height of 3 cm as same as the common size of containers used in container citrus nursery in China. The moving laser ranger was programmed to scan the soil surface after all motion of soil from one shock ceases (waiting time set to 0.5 s). Before drop shocks, we manually overfilled the container with soil material with a plastic measuring cup and kept shaking the cup to obtain a steady soil flow, and then scraped off the soil beyond the top tube rim with a stainless-steel ruler to make the soil initial volume always equal to the container capacity. After drop shocks, the overall packing density and its sequence were calculated from ranging values collected from the laser ranger and the soil mass measured on an electronic scale (CHS-D, range: 30 kg, resolution: 0.1g). With the cyclic operations of removing the top tube, scraping off soil beyond the top tube rim and weighing the container and soil inside it, we calculated the local packing density of compacted soil in each tube from differences of these weighting values (Fig 3). One should note that these operations are destructive, and increase significantly the probability of ambient humidity changing the moisture content of limited soil material.

The values of drop height $H$ included 2 mm, 4 mm, 5 mm and 6 mm, whose average intensities were 1.93 g, 2.86 g, 3.60 g and 3.99 g respectively. The maximum of drop times was 1000.

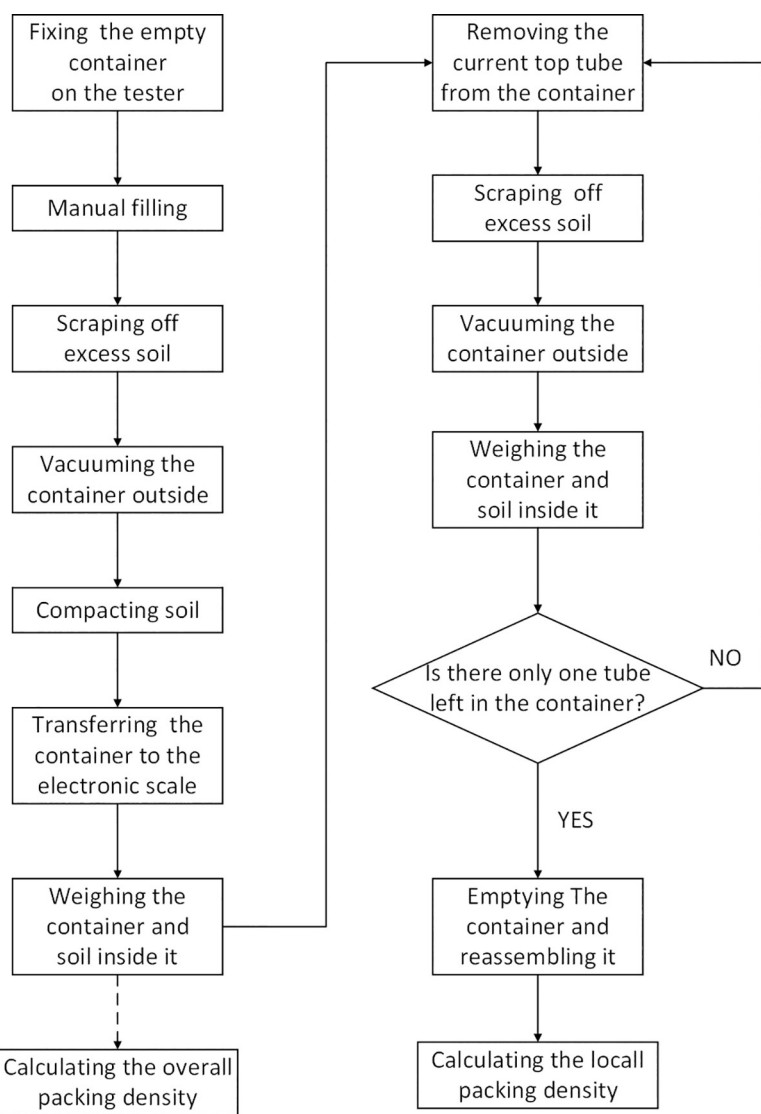

**Fig 3. The flow chart of experimental operations.** Manual filling means the container is overfilled slightly with soil using a plastic measuring cup, during which the cup need shaking to obtain a steady soil flow. Scraping off excess soil means carefully scraping soil beyond the rim of the top tube with a straight-edged thin plate to make soil volume consistent with the consistent the container capacity. Actually, a stainless-steel ruler was used. The dashed arrow indicates that the overall packing density could be calculated without further operations. The operation of remove-scrape-weigh refers to the operations forming a close loop.

The measurements were spaced out in drop times on a logarithmic scale with 200 measures of the overall packing density and 20 measurements of the local packing density. Operations with each setting repeatedly ran 3 times. The packed soil had the same initial volume of 3980 mL, and weighed 4457.1±38.4 g based on 200 filling operations (S1 Fig).

# Results and discussion

## Temporal evolution of soil packing density

Both the overall and local packing densities exhibited no sign of steady state and kept progressively increasing, due to the limited times of drop shocks (Fig 4). The packing densities at

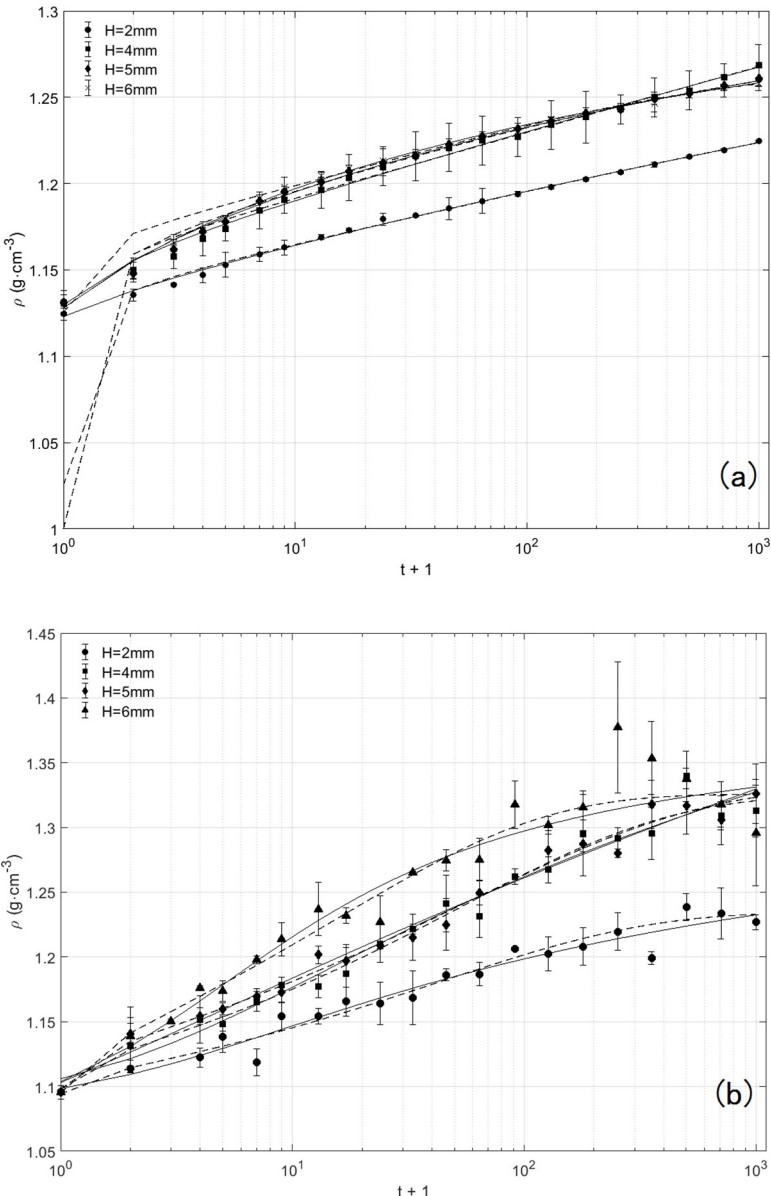

**Fig 4. Temporal evolution of packing density for various drop heights.** (a) Temporal evolution of the overall packing density: $H = 2$ mm (•), $H = 4$ mm (■), $H = 5$ mm (◆) and $H = 6$ mm (▲). Each curve is an average of 3 separate experimental runs and error bars represent the rms variation between runs. The solid lines are least square fits to the inverse-logarithmic function, namely Eq (1), and the dotted lines are fits to the stretched exponential function, namely Eq (2). Approximate 10% of the measurements are plotted for clarity (S1 File). (b) Temporal evolution of the local packing density obtained from the 8th tube from the bottom up at the same drop heights as (a) (S2 File).

$H = 2$ mm were significantly lower than those at other values of drop height. At $H = 4$ mm, 5 mm and 6 mm, the overall packing densities presented intersecting and overlapping. In despite of not reaching a hypothetical steady state, the increasing drop times changed gradually the positive correlation between the packing density and the drop height into a non-monotonic one (Fig 5). It indicates one could acquire the maximum steady-state packing density by adjusting the drop height $H$. According to the results available here, the overall packing density at $H = 4$ mm appears to be a peak value.

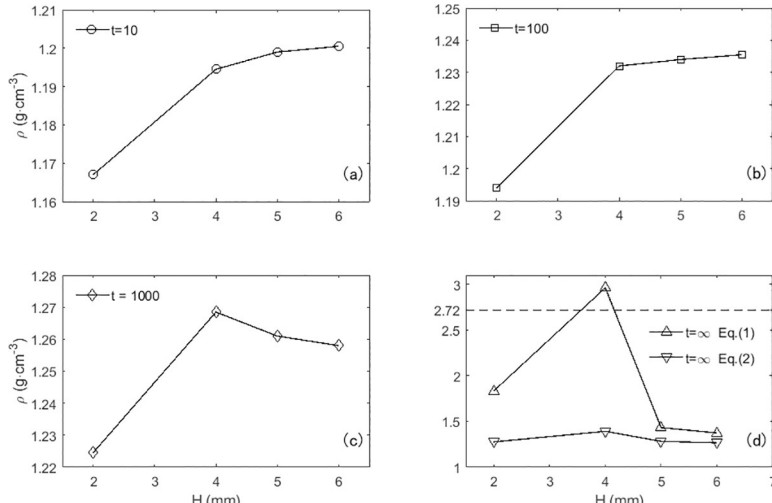

**Fig 5. The dependence of the overall packing density on the drop height.** Drop times: $t = 10$ (a), $t = 100$ (b), $t = 1000$ (c) and $t = \infty$ (d). The estimations of steady state ($t = \infty$) are obtained from the fits to the inverse-logarithmic function ($\triangle$) and the stretched exponential function ($\triangledown$). The dotted line in the panel (d) represents the relative density of soil.

The empirical inverse logarithmic formula [17] and the stretched exponential function [19] have been verified to be feasible and effective to represent the temporal evolution of the packing fraction of mono-disperse glass spheres under vertical tapping. The packing density measurements of poly-disperse soil here are analyzed separately through curve fitting to the two formulas (Fig 4):

$$\rho(t) = \rho_\infty - (\rho_\infty - \rho_o) \cdot \frac{1}{1 + Bln(1 + t/\tau_f)} \tag{1}$$

$$\rho(t) = \rho_\infty - (\rho_\infty - \rho_o) \cdot \exp(-(t/\tau_f)^\beta) \tag{2}$$

where the parameters $\rho_o$ and $\rho\infty$ are the packing densities of the initial state ($t = 0$) and the steady state ($t = \infty$) respectively. $\tau_f$ is a parameter purely related to the intensity $\Gamma$, and $\beta$ and $B$ are fitting parameters. The fitting results of overall packing densities are similarly satisfactory

**Table 1. The local packing density estimations of the steady-state.**

| tube | H = 2 mm | | H = 4 mm | | H = 5 mm | | H = 6 mm | |
|---|---|---|---|---|---|---|---|---|
| | Chicago | Rennes | Chicago | Rennes | Chicago | Rennes | Chicago | Rennes |
| 1 | 1.22 | 1.12 | 1.21 | 1.30 | 1.91 | 1.20 | 1.33 | 1.22 |
| 2 | 1.26 | 1.18 | 2.88 | 1.31 | 1.23 | 1.24 | 1.32 | 1.25 |
| 3 | 1.30 | 1.18 | 1.26 | 1.12 | 1.99 | 1.24 | 1.31 | 1.25 |
| 4 | 1.77 | 1.12 | - | 1.25 | 2.45 | 1.28 | 1.47 | 1.28 |
| 5 | 1.19 | 1.19 | - | 1.29 | 1.27 | 1.27 | 1.34 | 1.28 |
| 6 | 1.22 | 1.20 | - | 1.28 | 1.50 | 2.28 | 1.38 | 1.29 |
| 7 | - | 1.23 | - | 1.33 | 2.59 | 1.32 | 1.40 | 1.32 |
| 8 | - | 1.24 | - | 1.33 | - | 1.33 | 1.40 | 1.33 |

Chicago and Rennes are corresponding to Eq (1) and Eq (2), respectively. The index of each tube is numbered from the bottom up. The cells marked with "-" indicate tubes not filled due to the descent of soil free surface.

with R-square of 0.98~0.99 and RMSE of 0.0013~0.0023, except that the estimations of $\rho_o$ with Eq (2) deviates more from the measurements than those with Eq (1). But each estimation of $\rho_\infty$ with Eq (1) is greater than its counterpart with Eq (2), and even the value of overall packing density estimated by the inverse-logarithmic function at $H = 4$ mm is unreasonably greater than the relative density of soil particles (Fig 5D). There exist the same differences between the two equations with respect to the steady-state predictions of the local packing density (Table 1). In addition, the ratio of the container diameter (130 mm) to the average size of soil particles (0.86 mm) is great enough, neither producing strong boundary effects nor restraining soil's convection. The conclusion [20] on the steady state is verified by the comparison here that the inverse-logarithmic function applies to the compaction process of vibrated grains in slender containers. The compaction dynamics of soil here, therefore, should be represented by the stretched exponential function.

## The role of drop shocks in the compaction dynamics

As for the external mechanical excitations, soil compaction under drop shocks has some essential differences from the compaction of typical granular packing under vertical controlled tapping. The soil material is poly-disperse with rough and irregular particles, and it's inevitable for soil grains to crush or cohere during compaction, failing to keep a constant distribution of particle size (Fig 1A). This is beyond the restriction of so-called "soft" compaction [20]. Moreover, our experimental apparatus has been designed to apply vertical drop-induced taps to the soil packing in the container. The tap intensity is adjusted by just changing the drop height, rather than by monitoring the acceleration in real time and actively correcting any deviation from the set value as an electromagnetic vibrator with the closed-loop control strategy dose.

The intensity $\Gamma$ actually experienced a sharp decrease followed by an increase of slowing rate during once compaction, and reached a minimum at $t = 3$ regardless of the drop height (Fig 6). It may indicate that the energy absorbed by the soil packing increased for the first three drop shocks, and then decreased towards 0 for the other shocks with the soil packing

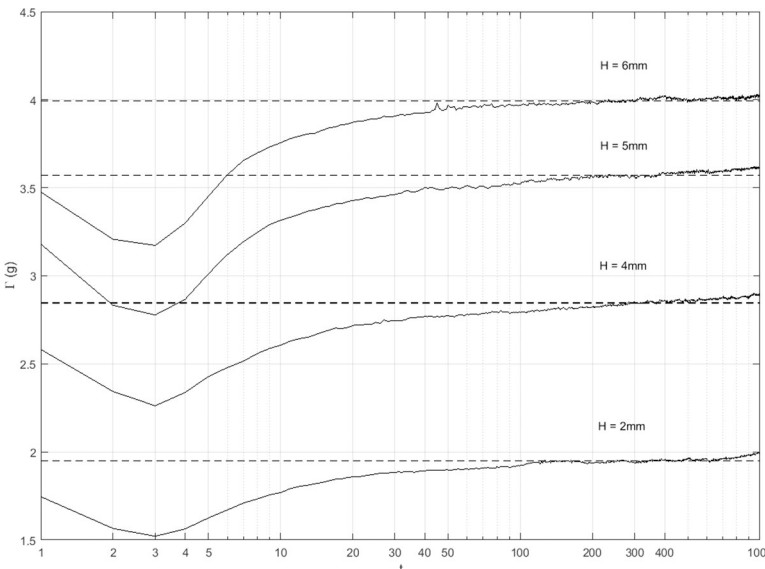

**Fig 6. The temporal evolution of the intensity.** The acceleration signal was measured by the accelerometer connected with the cylindrical container at the sampling frequency of 3.2 kHz. The raw data was low-pass filtered with the cutoff frequency of 50 Hz before the peak values were extracted.

approaching a "steady state". Soil got compacted under the condition of varying intensity, due to the interaction between the soil packing and the drop shocks produced by the combination of the cylindrical vessel and the soil. This is an inherent characteristic for the compaction method based on drop shocks. Here we try to confirm again the applicability of the stretched exponential function to the compaction dynamics under shocks of a constant drop height from the perspective of the intensity of drop shocks. Its parameter $\tau_f$ relates to the intensity according to the Arrhenius equation (19), namely,

$$\tau_f = \tau_o \cdot \exp\left(\frac{\Gamma_o}{\Gamma}\right) \tag{3}$$

where $\tau_o$ and $\Gamma_o$ are constants associated with experimental conditions. We convert the item containing $\tau_f$ into the form:

$$\frac{t}{\tau_f(t,H)} = \frac{1}{\tau_o} \cdot \frac{t}{\exp(\Gamma_o \cdot \Gamma^{-1}(t,H))} \tag{4}$$

Hence the intensity $\Gamma$ essentially is a bivariate function of the variables of drop times and drop height, with same trends at different values of drop height (Fig 6). However, Eq (4) appears as a straight line of $t$ regardless of the constant $\Gamma_o$, and its slop has a positive correlation with the drop height (Fig 7). This means that $\frac{1}{\tau_f(t,H)}$ in effect has nothing to do with $t$ and is corresponding to the drop height H. As a result, the condition of a constant drop height can be equivalently transferred to that of a constant intensity.

## Vertical distribution of soil packing density

The slow descent of soil free surface is the external manifestation of its internal packing state. The interaction between soil grains and between soil grain and the vessel sidewalls, producing some unique causing some unique properties, like "Janssen effect", are always preventing grains from flowing freely and filling evenly the container. We gained the soil mass in each tube of the container by the operation of remove-scrape-weigh (Fig 8). It revealed that the accumulative mass of soil increased linearly with the number of tubes except ones not filled with soil, so the local packing density of soil in the container varied synchronously and kept a uniform distribution on the whole.

In order to explain the temporal evolution and one-dimensional distribution of soil packing density in further, the coordinate plane composed of the drop times and the ratio of the height to the total height of the soil packing has been used to represent the accumulative packing density with contour plots (Fig 9, S3 File). Once subjected to drop shocks, the accumulated density of the soil packing revealed a change from the initial uniformity into a slight positive gradient, similar to local packing density distribution of the mono-disperse beads [17, 19]. The local packing density increased with the height, while the contours approach to the bottom of the vessel with the increasing drop times. It seems that the increasing trend spread from the top down with a slowing rate, and the drop height determined the initial rate.

## Conclusions

We have experimentally investigated the compaction dynamics of air-dried soil under drop shocks. It allowed some conclusions and also exposed deficiencies of this work. The dependence of the packing density on the drop times can be more reasonably described by the stretched exponential function than the inverse-logarithmic function. The intensity of drop shocks varies with the drop times due to the interaction between the drop shocks and the soil

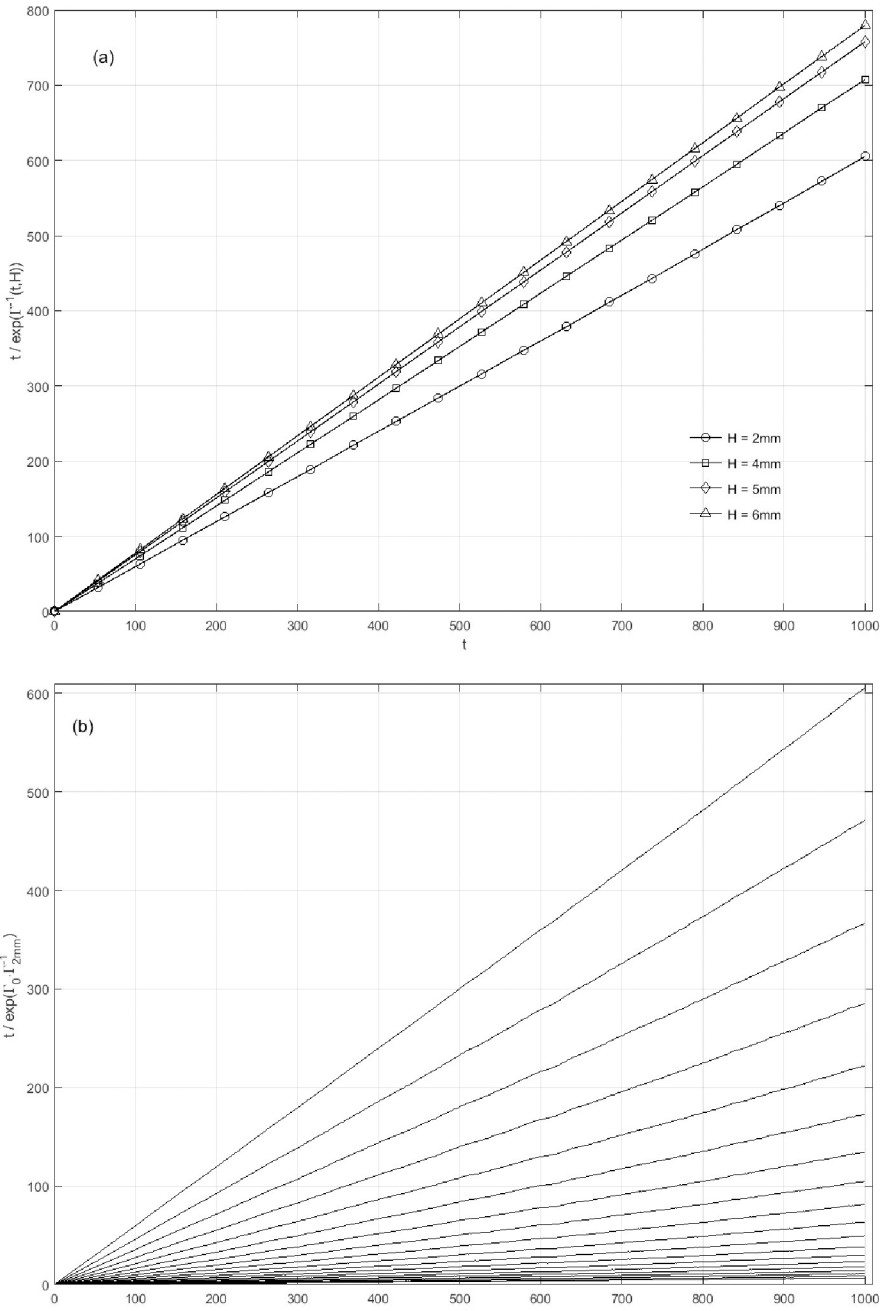

**Fig 7.** The dependence of $\frac{t}{\tau_f(t,H)}$ on the drop times $t$ and the drop height H. (a) The function of $\frac{t}{\exp(\Gamma^{-1}(t,H))}$ at different drop height H: $H = 2$ mm ($\circ$), $H = 4$ mm ($\square$), $H = 5$ mm ($\diamond$) and $H = 6$ mm ($\triangle$). Part of the data points are plotted for clarity. (b) The dependence of $\frac{t}{\exp(\Gamma_o \cdot \Gamma^{-1}(t,H))}$ on the $\Gamma_o$ at $H = 2$ mm, and $\Gamma_o$ ranges from 0.5 to 10 with the interval of 0.5. This cluster of curves illustrates that $\Gamma_o$ makes no difference to the linear relationship between $\frac{t}{\tau_f(t,H)}$ and t.

packing, despite this variation makes no difference to the applicability of the stretched exponential function for the reason that the compaction with a constant drop height can be converted equivalently into that with a constant intensity. The dependence of the packing density on the drop height is non-monotonic and there exists a threshold of the drop height

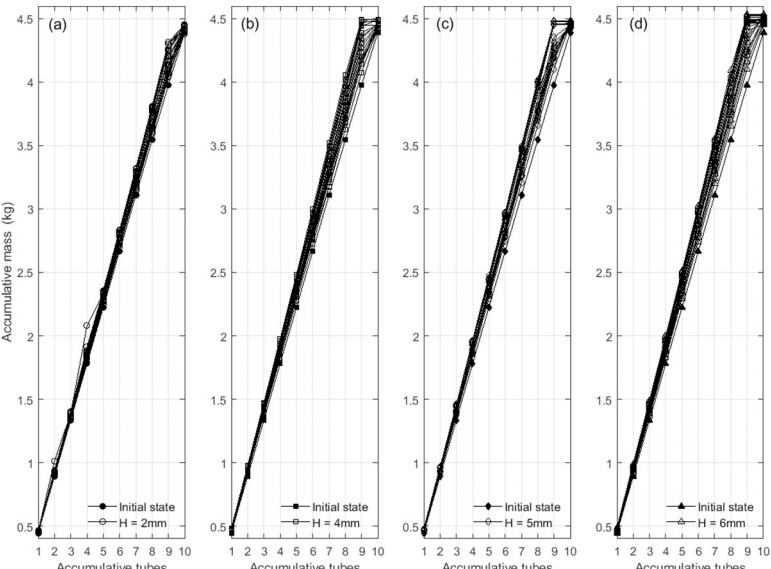

**Fig 8. The accumulative mass distributions of compacted soil at different drop height.** *H*: *H* = 2 mm (a), *H* = 4 mm (b), *H* = 5 mm (c) and *H* = 6 mm (d). The lines within each panel correspond to the 20 measurements of the local packing density.

corresponding to the maximum packing density. In addition to the temporal evolution, the accumulated packing density of air-dried soil appears a positive gradient along the height in agreement with the typical granular system composed of mono-disperse beads. But allowing for the complexity and diversity of growing media used in container nursery culture, it requires experiments and trials in further to focus on the influence of particle properties, such

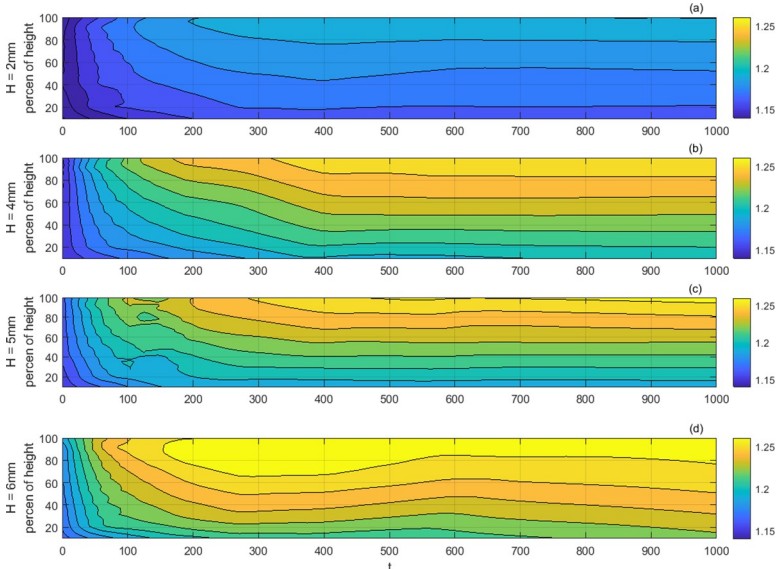

**Fig 9. The contour plots of the accumulative packing density at various drop height.** *H*: *H* = 2 mm (a), *H* = 4 mm (b), *H* = 5 mm (c) and *H* = 6 mm (d). Each panel here was obtained by the method of locally weighted linear regression, based on an average of 3 groups of separate measurements for the local packing density.

as the content of water and organic matter, on the compaction dynamics and the applicability of this compaction method.

## Supporting information

**S1 Fig. The histogram of soil initial packing density.**
(TIF)

**S1 File. The curve fittings of the overall packing density.**
(SFIT)

**S2 File. The curve fittings of the local packing density.**
(SFIT)

**S3 File. The contours of the local packing density.**
(SFIT)

## Acknowledgments

The authors thank the authorities of South China Agricultural University for permitting us to take the soil material from its campus.

## Author Contributions

**Conceptualization:** Jianbo Wang, Zhen Li.

**Methodology:** Tiansheng Hong.

**Supervision:** Jianbo Wang, Xiuyun Xue.

**Visualization:** Jianbo Wang, Xiuyun Xue.

**Writing – original draft:** Jianbo Wang.

**Writing – review & editing:** Jianbo Wang, Zhen Li, Shilei Lyu.

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
