## [Decision Letter · Decision Letter 0]

1 Feb 2021

PONE-D-20-39586

Experiment on compaction of air-dried soil under drop shocks

PLOS ONE

Dear Dr. Li,

Thank you for submitting your manuscript to PLOS ONE. After careful consideration, we feel that it has merit but does not fully meet PLOS ONE’s publication criteria as it currently stands. Therefore, we invite you to submit a revised version of the manuscript that addresses the points raised during the review process.

Please consider the comments of Reviewer 1 carefully as they are required for acceptance. I recommend that you also include some sentences in the manuscript that considers the position of Reviewer 2.  It will be more helpful if the authors replace most of the manuscript's dated references with more recent references (within the past ten years). 

We look forward to receiving your revised manuscript.

Kind regards,

Reginald B. Kogbara, Ph.D.

Academic Editor

PLOS ONE

Reviewers' comments:

Reviewer's Responses to Questions

**Comments to the Author**

1. Is the manuscript technically sound, and do the data support the conclusions?

Reviewer #1: Yes

Reviewer #2: Yes

2. Has the statistical analysis been performed appropriately and rigorously? 

Reviewer #1: Yes

Reviewer #2: Yes

3. Have the authors made all data underlying the findings in their manuscript fully available?

Reviewer #1: Yes

Reviewer #2: Yes

4. Is the manuscript presented in an intelligible fashion and written in standard English?

Reviewer #1: Yes

Reviewer #2: Yes

5. Review Comments to the Author

Reviewer #1: zand drop times were considered and the evolution and distribution of packing density were studied. The results showed that soil compaction caused by its drop shocks equals to that under the condition of controlled intensity; the distribution of local packing density showed a slight positive gradient. Although the findings in this study may improve container seedlings, this manuscript still need to be revised before possible publication.

1. In introduction, more references should be added, such as in line 34-36. What is the connection between these two paragraphs? Why use but in line 61, 66, and 67? What is the influence of local over compaction?

2. In material, add references to test methods such as oven drying, pycnometer method. Why show Figure 1 in 2.1 section where the experiment process has not been introduced. How does soil be compacted into container before drop shock? The experiment process is better shown in figure since the description is too long. Water content is a important factor during compaction, why is the effect of different water content is not considered during compaction?

3. How is soil density in each tube measured? Why does soil density in tube 8 (Figure 3b) finally decrease as the drop times increase? What do you mean “sinter” in line 189?

4. There are some grammar and format problems existing in the manuscript. The authors need to do a thorough check. Some examples are given as follow,

4.1. Line 44, “is” should be “has been”.

4.2. Line 52, “operators” should be “Operators”.

4.3. A space should be put between number and unit, such as 2 mm.

4.4. Figure 3 is not clear. Larger legends should be used.

Reviewer #2: The authors started an interesting work, but it still needs to more tests to be published. I suggest tests in field, where the variability of conditions, physical, chemical and biological soil, will alter the conclusions made.

6. PLOS authors have the option to publish the peer review history of their article (what does this mean?). If published, this will include your full peer review and any attached files.

Reviewer #1: No

Reviewer #2: No

---

## [Author Response · Author response to Decision Letter 0]

11 Mar 2021

Dear Dr. Kogbara and Reviewers,

Thank you very much for giving us an opportunity to revise our manuscript. We appreciate you for your comments and suggestions on our manuscript entitled “Experiment on compaction of air-dried soil under drop shocks” (PONE-D-20-39586).

We have studied reviewers’ comments carefully. According to the reviewers’ detailed suggestions, we have made a careful revision on the original manuscript. All revised portions are marked in red in the revised manuscript which we would like to submit for your kind consideration.

Kind regards,

Jianbo Wang

E-mail: jianbowang@stu.scau.edu.cn

Corresponding author: Zhen Li

E-mail: lizhen@scau.edu.cn

Responds to reviewers’ comments:

Reviewer #1

Comment 1: In introduction, more references should be added, such as in line 34-36. What is the connection between these two paragraphs? Why use but in line 61, 66, and 67? What is the influence of local over compaction?

Response: (1) We have added more references in Introduction (line42, 43, 46,76 and 77), and removed some out-of-date reference. Actually, some redundant sentences have been deleted. 

(2) We have changed Introduction into 3 paragraphs and added some sentences at the end of the 1st paragraph (line 54-61), to make clear the logical connection between the 1st and 2nd paragraphs. We have found the manual operation of compacting growing media by tapping containers on the table is significantly different from the mechanical compression method, and allows the synchronous executions of filling containers, compacting media and transplanting seedlings (or sowing). We have to investigate and understand the compaction dynamics of growing media under tapping or other mechanical disturbance base on the existing research results and conclusions related to the compaction of granular materials, in order to mechanize this compaction method.

(3) We misused the word “but” in line 61, 66 and 67 of the original manuscript, leading to a confusion in logic. In line 71 and 80 of the revised manuscript, we have corrected this mistake.

(4) Over compaction of growing media or agricultural soil usually deteriorate the cultural characteristics, say, leading to excessive penetration resistance to roots growth.

Here, we want to confirm whether there exists possible over compaction in the compacted growing media through the spatial distributions of local packing density. Local over compaction means a fairly uneven distribution of the packing density. We have added conferences in line 84.

Comment 2: In material, add references to test methods such as oven drying, pycnometer method. Why show Figure 1 in 2.1 section where the experiment process has not been introduced. How does soil be compacted into container before drop shock? The experiment process is better shown in figure since the description is too long. Water content is an important factor during compaction, why is the effect of different water content is not considered during compaction?

Response: (1) We have added conferences of testing method in line 96-98.

(2) It is really our fault that we didn’t introduce the experimental process. We have added sentences in line 128-133 to introduce the calculations of the overall and local packing density. And Fig 3 has been added to show the complete experimental process in the form of flow chart.

(3) We have added sentences in line 125-128 and the legend of Fig 3 to describe the filling method before drop shocks. We used the manual overfilling operation of pouring soil material into the container with a measuring cup, during which the cup needs shaking to obtain a steady soil flow. And then soil beyond the container rim would be scraped off with a stainless-steel ruler to make a constant initial volume of soil. 

(4) Indeed, the moisture content of soil will significantly change its compaction dynamics. We have added related sentences and references in line 76-80. The reason for not considering the moisture content is that, although the operation of remove-scrape-weigh was a direct method to measure the local packing density, it's very cumbersome and time consuming and greatly increases the frequency of soil contact with air, making it difficult for soil to keep the constant moisture content. That’s why we chose the air-dried soil material. Related sentences have been added in line 93,94,133 and134. We have to consider an indirect method to measure the local pacing density of wet soil, like testing penetration resistance.

Comment 3: How is soil density in each tube measured? Why does soil density in tube 8 (Figure 3b) finally decrease as the drop times increase? What do you mean “sinter” in line 189?

Response: (1) Soil density in each tube, also called the local packing density, is the soil mass in the tube divided by its volume. the soil volume in filled tubes is known, while the soil volume in unfilled tubes is calculated from the ranging values. Soil mass is the weighing values difference between two adjacent operations of remove-scrape-weigh. Related sentences have been added in line 130-133 and Fig 3 legend.

(2) The soil density decrease with drop times were cause by the instability (or randomness) of the two manual operations, including the filling operation and the operation of remove-scrape-weigh, because it can be observed (in Fig 4 of the revised manuscript) that the error bars of these points were prominently longer than other error bars.

(3) The word “sinter” was a mistake, and we have replaced it with “cohere” in line 213

Comment 4: There are some grammar and format problems existing in the manuscript. The authors need to do a thorough check. Some examples are given as follow,

4.1. Line 44, “is” should be “has been”.

4.2. Line 52, “operators” should be “Operators”.

4.3. A space should be put between number and unit, such as 2 mm.

4.4. Figure 3 is not clear. Larger legends should be used.

Response: It’s so kind of you to point out these grammar and format errors for us. We have checked the revised manuscript.

Reviewer #2

Comment 1: The authors started an interesting work, but it still needs to more tests to be published. I suggest tests in field, where the variability of conditions, physical, chemical and biological soil, will alter the conclusions made.

Response: Thank you for this comment. Indeed, any growing medium or agricultural soil is a complex mixture with the physical, chemical and biological properties. These properties have significant influence on the compaction dynamics under vertical tapping. We have added sentences and conferences in line 76-81. In this work, with the final purpose of mechanization of compacting growing media by tapping containers, we experimentally investigated the temporal evolution and vertical distribution of the soil packing density under drop shocks, as the basis for further research. Considering many factors at once maybe cause confusion in understanding, and concern about time and cost. We have added related sentences in line 288-291.

---

## [Decision Letter · Decision Letter 1]

31 Mar 2021

Experiment on compaction of air-dried soil under drop shocks

PONE-D-20-39586R1

Dear Dr. Li,

We’re pleased to inform you that your manuscript has been judged scientifically suitable for publication and will be formally accepted for publication once it meets all outstanding technical requirements.

Kind regards,

Reginald B. Kogbara, Ph.D.

Academic Editor

PLOS ONE

Additional Editor Comments (optional):

Reviewers' comments:

Reviewer's Responses to Questions

**Comments to the Author**

1. If the authors have adequately addressed your comments raised in a previous round of review and you feel that this manuscript is now acceptable for publication, you may indicate that here to bypass the “Comments to the Author” section, enter your conflict of interest statement in the “Confidential to Editor” section, and submit your "Accept" recommendation.

Reviewer #1: All comments have been addressed

Reviewer #2: All comments have been addressed

2. Is the manuscript technically sound, and do the data support the conclusions?

Reviewer #1: Yes

Reviewer #2: Yes

3. Has the statistical analysis been performed appropriately and rigorously? 

Reviewer #1: N/A

Reviewer #2: Yes

4. Have the authors made all data underlying the findings in their manuscript fully available?

Reviewer #1: Yes

Reviewer #2: Yes

5. Is the manuscript presented in an intelligible fashion and written in standard English?

Reviewer #1: Yes

Reviewer #2: Yes

6. Review Comments to the Author

Reviewer #1: The authors have addressed the comments, and hence I recommend that the manuscript be accepted to publish.

Reviewer #2: I would like to see the rest of the data for a complete understanding of the compaction process. However, I believe that the current text will be of great use to the scientific community.

7. PLOS authors have the option to publish the peer review history of their article (what does this mean?). If published, this will include your full peer review and any attached files.

Reviewer #1: No

Reviewer #2: No

---

## [Editor Report · Acceptance letter]

5 Apr 2021

PONE-D-20-39586R1 

Experiment on compaction of air-dried soil under drop shocks 

Dear Dr. Li:

I'm pleased to inform you that your manuscript has been deemed suitable for publication in PLOS ONE. Congratulations! Your manuscript is now with our production department. 

Kind regards, 

on behalf of

Dr. Reginald B. Kogbara 

Academic Editor

PLOS ONE